# SmartCrawler: A Size-Adaptable In-Pipe Wireless Robotic System with Two-Phase Motion Control Algorithm in Water Distribution Systems [note 1]

**DOI:** 10.3390/s22249666

**Published:** 2022-12-09

**Authors:** Saber Kazeminasab, M. Katherine Banks

**Affiliations:** 1Postdoctoral Research Fellow, Harvard University, Cambridge, MA 02138, USA; 2Zachry Department of Civil and Environmental Engineering, Texas A & M University, College Station, TX 77843, USA

**Keywords:** in-pipe robots, multi-phase motion controller, wireless sensor module, wireless control

## Abstract

Incidents to pipes cause damage in water distribution systems (WDS) and access to all parts of the WDS is a challenging task. In this paper, we propose an integrated wireless robotic system for in-pipe missions that includes an agile, maneuverable, and size-adaptable (9-in to 22-in) in-pipe robot, “SmartCrawler”, with 1.56 m/s maximum speed. We develop a two-phase motion control algorithm that enables reliable motion in straight and rotation in non-straight configurations of in-service WDS. We also propose a bi-directional wireless sensor module based on active radio frequency identification (RFID) working in 434 MHz carrier frequency and 120 kbps for up to 5 sensor measurements to enable wireless underground communication with the burial depth of 1.5 m. The integration of the proposed wireless sensor module and the two-phase motion controller demonstrates promising results for wireless control of the in-pipe robot and multi-parameter sensor transmission for in-pipe missions.

## 1. Introduction

Water Distribution Systems are crucial infrastructures in the water industry to deliver fluid to different cities and residential areas. Incidents to the pipelines cause leaks and water loss. The overall water pipe breaks has increased around 27% from 2012 to 2018 in the U.S. and Canada [1]. In addition, the parameters in water distribution systems need to be regularly monitored to ensure the quality of water that is vital to public health [2]. Due to the passive motion of mobile sensors [3] and complicated configuration of pipelines, they are not directed to the desired location, unless that part of the network that is considered for sensor operation is isolated from the rest of the network. Hence, it is required to design the sensor modules with independent motion of the flow. In-pipe robots are promising alternatives for these tasks, as they can embed various sensor modules for doing different in-pipe missions.

## 2. Related Work

### 2.1. Mechanisms of In-Pipe Robots

The mechanisms of the in-pipe robots are categorized into active and passive locomotion. The active locomotion robots are divided into wheeled type, Caterpillar type, and non-wheeled type mechanisms. The wheeled robots are divided into three categories of simple structure, wall-press, and screw-type robots. In the simple structure robots, wheels are connected under the body of the robot and move inside the pipe without anchoring to the pipe wall [4]. In the wall-press robots, however, the wheels anchor on the pipe wall and press it. The wheels are usually connected at the end of adaptable arms, on which these arms are anchored, on a body with a 120${}^{\circ }$ angle [5] (Our system). In the third types of wheeled robots (screw drive types), the spiral motion of the wheels is transmitted to linear through the wheels and are connected on a rotational and fixed units [6]. In the Caterpillar-type robots, the belt mechanism moves the robot [7]. The robots that are not powered by wheels can be divided into snake, inchworm, legged, and free-swimming types. In the snake types, some modular units are connected and move the robot inside the pipe [8]. Contraction and extension of a flexible body is the power transmission mechanism for the Inchworm types [9]. The legged robots are less common for in-pipe missions; however, they are more maneuverable (compared to other types), and they need more complex motion controllers [10]. The free-swimming robots move with propellers without connection to the pipe wall [11]. The passive locomotion robots use pressure inspection gauge (PIG) mechanisms [2].

Most robots are actuated by pneumatic [12] or electrical actuators [13]. There are some considerations in the design of in-pipe robots that need to be addressed: size-adaptability, and independent power supply for the long-distance travel of the robots in the pipeline. The short length of cables limits their long pipe inspection for cable-operated robots [14], which is not the case for battery-operated robots [7]. Moreover, there are disturbances in *in-service* distribution networks and the in-pipe robots are likely to collapse during operation, requiring reliable motion controllers [15].

### 2.2. Communication with Based Station

In cable-operated robots, data between robot and base station (BS) are transmitted through wired communication. Hence, for long-distance inspection, wireless communication is required for in-pipe robots. However, wireless communication through harsh and dynamic environment of soil and water is a challenging task [16], due to high signal attenuation in the underground environments and highly dynamic communication channels. To mitigate these challenges, researchers employ EM waves with supplementary mechanisms [17]. However, this method is limited to straight pipes.

### 2.3. Localization and Navigation of In-Pipe Robots

In addition, it is required to localize the in-pipe robots in the network, since they move in long and complicated configurations of pipelines [18]. Moreover, autonomous navigation of in-pipe robots toward long-distance inspection is highly desirable, and recent works have tried to address this challenge [19]. In [19], negotiation of the robot in non-straight configurations with small curvature is addressed; however, configurations such as 90-degree bends and T-junctions that are more prevalent in distribution systems remain challenging, specifically T-junctions [20,21].

### 2.4. Technical Gap

Lack of a fully automotive in-pipe robot that can inspect the long distance in highly pressurized **in-service** WDS.Lack of reliable motion controller for in-pipe robots in straight and non-straight configurations of pipeline networks.A bi-directional wireless sensor module is required for the wireless access to the underground robot.

In this paper, we design and develop an in-pipe robotic system that is equipped with a motion controller and wireless sensor module. Our contribution can be summarized as:We design a self-powered size-adaptable in-pipe robot with high agility and maneuverability. The components of the robot are customized considering flow condition.We propose a two-phase motion controller for the reliable motion of the robot in straight and non-straight configurations of pipelines.We propose a wireless sensor module that facilitates reliable communication link with high data throughput and customized antenna in the confined space of the robot.We synchronize the two-phase motion controller and the wireless sensor module to develop the wireless controller and multi-receiver wireless network for the in-pipe robot.

## 3. SmartCrawler Design

The SmartCralwer comprises one central processor unit, three adjustable arm modules, and three actuator modules. The arm modules are anchored on the central processor with 120° angle. At the end of arm modules, actuator modules are connected that move the robot inside the pipe by gear motors. The robot benefits from the flexibility characteristic of soft robots and dexterity and mobility of motorized robots (see Figure 1). We characterized the robot’s components for work in in-service distribution systems and addressed the health considerations in our previous work in [5]. The general parameters of the robot are listed in Table 1.

In following, we provide more details on each part of the robot.

### 3.1. Central Processor Unit

The central processor comprises two parts: one part locates the sensor modules and the other part locates the embedded system in the central processor. These parts are fixed together with a seal mechanism. The central processor hosts the sensor module, embedded system, and the battery (see Figure 2).

#### 3.1.1. Arm Module

We designed adjustable arm modules with three Boomerang-shaped arms and passive springs that are connected to the central processor (see Figure 3).

#### 3.1.2. Actuator Module

At the end of each arm module, an actuator module is connected that includes a gear motor, a motor cover, a rubber wheel, and a pair of ball bearings (see Figure 4).

The robot moves inside the pipe by actuator modules that are connected at the end of the arm modules. A passive spring is located between each arm and the central processor (i.e., one end is anchored on the arm and the other end is anchored on the central processor) and provides the “press force” for the wheel and also makes the outer size of the robot adjustable to the pipe size.

For the control system observer sensor, an inertial measurement unit (IMU) and motor encoders are needed. These sensors are chosen from the category of sensors for precise applications. BMI160 (i.e., IMU), is a low-noise 16-bit IMU designed for mobile applications, providing highly accurate sensor data and real-time sensor data (16,384 LSB/g for gyro and 262.4 LSB/${}^{\circ }$/s for accelerometer). Moreover, the motor encoders are six channels ENX-EASY-16 with 1024 pulse/turn, and the gear motor of 26:1 ratio. Hence, the accuracy of the encoder is 0.013${}^{\circ }$ (26,624 pulses/turn). Figure 5a shows the hardware architecture of the PCB and includes a power management circuit, analog drivers for the gear motors, analog sensors, and a low-power embedded system. Moreoverr, Figure 5b shows the PCB and the assembly of the PCB and the central processor.

### 3.2. Spring Characterization

To have a reliable motion for the robot, the pure rolling of the wheels is important and the friction force between the pipe wall and the wheels should be enough to this aim. The passive springs press the wheels and provide friction force ($F_{N}$ and $F_{N}^{\prime }$ in Figure 6a). The weight of the robot adds to the normal force for the wheels below the center of mass of the robot (i.e., $F_{N}$); hence, the required spring stiffness for the associated wheels is less than the spring stiffness that provides $F_{N}^{\prime }$. In this regard, we provide our analysis for $F_{N}^{\prime }$:(1)$$\left(mg-F_{N}^{\prime }\right)a\cos\left(\beta \right)-f_{s}H+F_{Spring}X_{Spring}=0$$

In ***OAB***, and in Figure 6b:(2)$$\beta =\alpha +(\frac{\pi }{2}-\theta )$$
(3)$$\alpha =\arcsin(\frac{t}{a}\cos\left(\theta \right))$$

Plugging (3) in (2):(4)$$\beta =-\theta +\arcsin\left(\frac{t}{a}\cos\left(\theta \right)\right)+\frac{\pi }{2}$$

Also, from Figure 6b, we have:(5)$$\beta =\arcsin(\frac{H}{L_{a}})$$
where $L_{a}$ is the arm length. Plugging (5) in (4):(6)$$\arcsin\left(\frac{H}{L_{a}}\right)=-\theta +\arcsin\left(\frac{t}{a}\cos\left(\theta \right)\right)+\frac{\pi }{2}$$

From (6), we know θ=g(H) and the size of the pipe determines θ. We also have:(7)$$X_{Spring}=t\cos\left(\theta \right)$$

Plugging (7) in (1):(8)$$F_{Spring}=\frac{1}{t\cos(g(H))}\left(\left(F_{N}^{\prime }-mg\right)a\cos\left(\arcsin\left(\frac{H}{L_{a}}\right)\right)-f_{s}H\right)$$

As the springs are linear, we have $F_{Spring}=K_{Spring}L_{disp}$. We have:(9)$$L_{disp}=\sqrt{{\left(t+\cos\left(arcsin\left(\frac{H}{L_{a}}\right)\right)\right)}^{2}+{\left(a\arcsin\left(\frac{H}{L_{a}}\right)\right)}^{2}}\left(1-\cos\left(g\left(H\right)\right)\right)=KU\left(H\right)$$

From (8) and (9):(10)$$K_{Spring}=\frac{1}{tU(H)\cos(g(H))}\left(\left(F_{N}^{\prime }-mg\right)a\cos\left(\arcsin\left(\frac{H}{L_{a}}\right)\right)-f_{s}H\right)=G\left(H\right)$$

To account for all situations, we would need:(11)K=max(G(H))

Moreover, the friction between the pipe wall and the wheels is considered Coulomb friction:(12)$$f_{s(max)}=\mu_{s}F_{N}^{\prime }$$

The $\mu_{s}$ is the static friction coefficient and considered 0.8 for the pipe wall and wheel contact. The maximum static force should be always higher than the drag force applied on the wheels. To calculate the drag force, we performed a computational fluid dynamic (CFD) work [5] (Figure 6c) and considered an extreme scenario in which the robot in its extended size (i.e., 22-in) moves with 50 m/s velocity in the opposite direction of a flow with 70 m/s. In this scenario, the drag is ≈26 N, and due to the symmetry of the robot geometry, the drag force for each wheel is ≈8.7 N. based on (12), $F_{N}^{\prime }\approx 11$ N. Considering (11), we calculated the required spring stiffness for each *H* (Figure 6d) and selected the maximum value (i.e., ≈ 3.36 kN/m).

The selected spring stiffness ensures **pure rolling** for the wheels in all situations of the robot operation in the presence of water flow. Pure rolling is vital for the development of a robust motion controller for the robot working in disturbing environment.

### 3.3. Modeling

The free-body diagram of the robot in a pipe is shown in Figure 7, and the parameters are described in Table 2. The equations of the motion of the robot are presented as follows:(13)$$\frac{\tau_{1}}{R}+\frac{\tau_{2}}{R}+\frac{\tau_{3}}{R}+\frac{1}{2}\rho AC_{d}{\left(\dot{x}-V_{f}\right)}^{2}=m\ddot{x}$$
(14)$$\frac{1}{2}\frac{\tau_{3}}{R}L_{a}\cos\left(\theta_{3}-\phi \right)-\frac{1}{2}\frac{\tau_{2}}{R}L_{a}\cos\left(\theta_{2}+\phi \right)=I_{yy}\ddot{\phi }$$
(15)$$\begin{array}{c} \begin{array}{c} \frac{\sqrt{3}}{2}\frac{\tau_{3}}{R}L_{a}\cos\left(\theta_{3}-\phi \right)\left(1+\sin\psi \right)+\\ \frac{\sqrt{3}}{2}\frac{\tau_{2}}{R}L_{a}\cos\left(\theta_{2}+\phi \right)\left(1+\sin\psi \right)-\\ \frac{\tau_{1}}{R}L_{a}\cos\left(\theta_{3}+\psi \right)-\\ mgL_{a}\sin\left(\theta_{1}+\psi \right)=I_{zz}\ddot{\psi } \end{array} \end{array}$$

## 4. Two-Phase Motion Controller

The robot experiences a change in pipe diameter, and there is high velocity flow that applies disturbance on the robot. The robot needs to track a desired velocity in pipelines and steer to the desired directions at non-straight configurations such as 90-degree bends and T-junctions. To this aim, we develop a two-phase control algorithm that a stabilizer–velocity tracker controller and a differential motion controller are proposed for straight and non-straight paths, respectively.

### 4.1. Phase 1: Stabilizer–Velocity Tracker

#### 4.1.1. LQR Stabilizer Controller

In the stabilization of the SmartCrawler, our goal is to locate the central processor at the center of the pipe during operation. To this aim, we consider $x_{s}={\left[\phi ,\dot{\phi },\psi ,\dot{\psi }\right]}^{T}$ as stabilizing states (see Figure 8) that need to be kept at zero during operation. We decoupled the dynamic equations of the system into two sets: One set is related to the robot’s linear motion, (Equation 13), and the other set is related to the orientation of the robot, (Equation 14) and (Equation 15). We then linearized (Equation 14) and (Equation 15) around the equilibrium point (i.e., $x_{s}^{e}={\left[0,0,0,0\right]}^{T}$) and derived the system’s auxiliary matrices as:
(16)$$x_{s}=\left[\begin{array}{c} \phi \\ \dot{\phi }\\ \psi \\ \dot{\psi } \end{array}\right]=\left[\begin{array}{c} x_{1}\\ x_{2}\\ x_{3}\\ x_{4} \end{array}\right]$$
(17)$$x_{s}=\left[\begin{array}{c} u_{1}\\ u_{2}\\ u_{3} \end{array}\right]=\left[\begin{array}{c} \tau_{1}\\ \tau_{2}\\ \tau_{3} \end{array}\right]$$
(18)$$\dot{x_{s}}=F\left(\theta_{1},\theta_{2},\theta_{3},\phi ,\psi ,u\right)=\left[\begin{array}{c} F_{1}\\ F_{2}\\ F_{3}\\ F_{4} \end{array}\right]$$
where $F_{1}=x_{2}$, $F_{2}=\frac{1}{RI_{yy}}\left[\frac{\sqrt{3}}{2}\tau_{3}L_{a}\cos\left(\theta_{3}-\phi \right)-\frac{\sqrt{3}}{2}\tau_{2}L_{a}\cos\left(\theta_{2}+\phi \right)\right]$, $F_{3}=x_{4}$, and $F_{4}=\frac{1}{I_{zz}}\left[\frac{1}{2R}\tau_{3}L_{a}\cos\left(\theta_{3}-\phi \right)+\frac{1}{2R}\tau_{2}L_{a}\cos\left(\theta_{2}+\phi \right)-\frac{1}{R}\tau_{1}L_{a}\cos\left(\theta_{1}+\psi \right)-mg\sin\left(\theta_{1}+\psi \right)\right]$. We have:(19)$$A_{s}=\left[\begin{array}{cccc} \frac{\partial F_{1}}{\partial x_{1}} & \frac{\partial F_{1}}{\partial x_{2}} & \frac{\partial F_{1}}{\partial x_{3}} & \frac{\partial F_{1}}{\partial x_{4}}\\ \frac{\partial F_{2}}{\partial x_{1}} & \frac{\partial F_{2}}{\partial x_{2}} & \frac{\partial F_{2}}{\partial x_{3}} & \frac{\partial F_{2}}{\partial x_{4}}\\ \frac{\partial F_{3}}{\partial x_{1}} & \frac{\partial F_{3}}{\partial x_{2}} & \frac{\partial F_{3}}{\partial x_{3}} & \frac{\partial F_{3}}{\partial x_{4}}\\ \frac{\partial F_{4}}{\partial x_{1}} & \frac{\partial F_{4}}{\partial x_{2}} & \frac{\partial F_{4}}{\partial x_{3}} & \frac{\partial F_{4}}{\partial x_{4}} \end{array}\right]$$
(20)$$B_{s}=\left[\begin{array}{ccc} \frac{\partial F_{1}}{\partial u_{1}} & \frac{\partial F_{1}}{\partial u_{2}} & \frac{\partial F_{1}}{\partial u_{3}}\\ \frac{\partial F_{2}}{\partial u_{1}} & \frac{\partial F_{2}}{\partial u_{2}} & \frac{\partial F_{2}}{\partial u_{3}}\\ \frac{\partial F_{3}}{\partial u_{1}} & \frac{\partial F_{3}}{\partial u_{2}} & \frac{\partial F_{3}}{\partial u_{3}}\\ \frac{\partial F_{4}}{\partial u_{1}} & \frac{\partial F_{4}}{\partial u_{2}} & \frac{\partial F_{4}}{\partial u_{3}} \end{array}\right]$$

Auxiliary matrices are calculated by substituting $x_{s}^{e}$ and $u_{e}$ in and, where $u_{e}$ is the controller input at the equilibrium point.
$$C_{s}=\left[\begin{array}{cccc} 0 & 1 & 0 & 0\\ 0 & 0 & 1 & 0 \end{array}\right]$$

Is the system’s output matrix. The system’s auxiliary representation in state-space form is:(21)$$\begin{array}{c} \dot{x_{s}}=A_{s}x_{s}+B_{s}u\\ y_{s}=C_{s}x_{s} \end{array}$$

Following this, we designed a state feedback controller based on a linear quadratic regulator (LQR). To this aim, we define a cost function:(22)$$J\left(K\right)=\int_{0}^{\infty }\left[x_{s}^{T}Qx_{s}+u^{T}Ru\right]dt$$
where *Q* weights $x_{s}$ and *R* weights *u*. *J* (*K*) is minimized once *K* is calculated as:(23)$$K=R^{-1}B_{s}^{T}P$$

*P* in (Equation 23) is calculated with algebraic Riccati equation:(24)$$-PA_{s}-A_{s}^{T}P-Q+PB_{s}R^{-1}B_{s}^{T}P=0$$

Finally, the output of LQR stabilizer is calculated as:(25)$$u=-Kx_{s}$$

The proposed LQR controller keeps $x_{s}$ at the equilibrium point, which results in the central processor remaining in the center of the pipe during operation.

#### 4.1.2. Velocity Controller

In addition to stabilization, the robot needs to move independently of the water flow with desired velocity. To this aim, we design a velocity controller. The angular velocity of the wheels of the robot is equal in the straight path; hence, the relation between the linear velocity of the robot and the encoder output is calculated as:(26)$$v=\frac{2\pi R}{NT_{c}}$$
where *N*, *R*, and $T_{c}$ are the number of pulses per wheel turn that the motors’ encoder generate, wheel radius, and the time between two consecutive pulses, respectively. In the velocity controller, the desired velocity is translated to the desired angular velocity, $\omega_{d}$, based on (Equation 26), and the three PID regulators track $\omega_{d}$.

The commands from PID controllers are amplified with L298N to drive the motors. The combined LQR stabilizer and velocity controller shapes the stabilizer–velocity controller, and is shown in Figure 9.

Controller performance evaluation with experimental results:

We developed a test-bed to evaluate the performance of the stabilizer–velocity controller (see Figure 10a). We used a super lithium-ion battery to power the robot. Since the IMU outputs are noisy and have a DC baseline, we used Mahony complementary filter [22], which is an optimal sensor fusion algorithm to compute the rotational angles of the robot. We considered four iterations for performance evaluation with different non-zero initial values for ϕ (i.e., $\phi_{0}$), ψ (i.e., $\psi_{0}$), and desired linear velocity in each iteration.

In iteration 1 (blue curve in Figure 10), $\phi_{0}=-4^{\circ }$ and $\psi_{0}=-3^{\circ }$. They converge to zero and fluctuate with the $\pm 2^{\circ }$ margin within two seconds. The fluctuation margins are the same for other iterations as well. The desired linear velocity in this iteration is 0.1 m/s and the robot reaches the desired velocity in around two seconds. In iteration 2 (red curve in Figure 10), $\phi_{0}=-14^{\circ }$ and $\psi_{0}=-11^{\circ }$, and the desired linear velocity is 0.2 m/s. The stabilizing states converge to zero in two seconds and the robot reaches 0.2 m/s in less than three seconds. In iteration 3 (orange line in Figure 10), $\phi_{0}=-9^{\circ }$ and $\psi_{0}=+5^{\circ }$. The robot reaches the desired linear velocity (i.e., 0.3 m/s) in four seconds and stabilizes in one second. The stabilization duration is around one second in iteration 4 (purple line in Figure 10) and the time the robot takes to reach the desired linear velocity of 0.35 m/s is five seconds. The experimental results prove that the developed stabilizer–velocity tracker controller can control the velocity of the underactuated robot and stabilize it in the uncertain environment of pipelines.

### 4.2. Phase 2: Controller Algorithm for Non-Straight Paths

The controller in this phase comprises a trajectory generator, an observer, an error-check sub-module, and three PID regulators. The trajectory generator creates differential motion based on the actuator architecture. Different angular velocities of the wheels change the orientation of the robot (see phase 2 block in Figure 11).

We call this algorithm the variable velocity allocation (VVA) and the maximum and minimum velocities are calculated based on the desired speed that guarantees smooth motion. We reached the maximum velocity of 60 rpm, and the minimum velocity of 30 rpm in our simulations in ADAMS, which is a dynamic simulation software to ensure a fast and smooth motion at the junctions. The observer of the controller includes the IMU and the motors’ encoders. Three PID regulators track the desired velocities. An error-check sub-module is designed to check the *status of the rotation* and allows the controller in this phase to continue until the desired rotation around the desired axis is acquired. Once the desired rotation is completed, the controller switches to phase 1.

Controller performance evaluation at 90-degree bend:

We did co-simulation in ADAMS and MATLAB to evaluate the performance of the proposed controller in the 90${}^{\circ }$ bend and T-junctions. Figure 12a shows the robot in a 12-in bent pipe. The front path is a 90${}^{\circ }$ bend in which the robot needs to rotate 90${}^{\circ }$ clockwise around the Y-axis (i.e., $\phi_{d}$ = −90${}^{\circ }$). The sequences of motion during the simulation is shown in Figure 12a and the robot’s angular velocities are shown in Figure 12b. The blue curve is the angular velocity around the y-axis that ranges between −10${}^{\circ }$/s and −25${}^{\circ }$/s. The green curve shows the angular velocity of the robot around the X-axis that ranges between +1${}^{\circ }$/s to −5${}^{\circ }$/s. The red curve shows the robot’s angular velocity around the Z-axis that ranges ±1${}^{\circ }$/s during motion. The angular velocities show that the robot rotates −90${}^{\circ }$ around the Y-axis, −8${}^{\circ }$ around the X-axis, and ≈0${}^{\circ }$ around the Z-axis. The rotation angles around each axis show the robot can pass bends smoothly. We repeated our simulations in different pipe diameters and validated that the robot can pass the bends with diameters range from 9-in to 22-in with the motion controller.

Controller performance evaluation in T-junctions:

The robot needs to rotate 90${}^{\circ }$ counterclockwise about the Y-axis (i.e., $\phi_{d}$ = +90${}^{\circ }$ ) in a 12-in T-junction, as shown in Figure 12c. The robot starts moving and the wheel that is further to the junction curvature, loses contact with *Pipe (I)*, and due to the pretension in springs, contact with the *Pipe (II)*. Other wheels smoothly enter *Pipe (II)*. The robot’s angular velocities and linear speed are shown in Figure 12d (the right vertical axis shows the linear speed and the left vertical axis shows angular velocities). The angular velocity around Y-axis (blue curve) ranges between +25${}^{\circ }$/s and +75${}^{\circ }$/s most of the time. However, in a short period, the velocity reaches +200${}^{\circ }$/s and −50${}^{\circ }$/s. The angular velocity around the x-axis (green curve) ranges between −37${}^{\circ }$/s and +25${}^{\circ }$/s during motion. The angular velocity around the z-axis (red curve) ranges between −50${}^{\circ }$/s to +100${}^{\circ }$/s and most of the time is positive. At the end, the rotation around the Z-axis is not near zero, as shown in Figure 12c(6), and is cancelled when the controller switches to phase 1. The dotted blue curve shows the robot’s linear speed that starts from zero and range between 0.15 m/s and 0.25 m/s. However, it is constant during most of the rotation that results in smooth rotation. The simulation results show that the proposed controller at phase 2 enables the robot to change its direction in the challenging environment of the T-junctions, as there is minimum available contact space for the wheels. We repeated our simulations in T-junctions with different diameters and validated that the robot can pass T-junctions with diameters ranging from 9-in to ≈15-in.

## 5. Wireless Sensor Module for SmartCrawler

In our application, the transceiver(s) is located outside the pipeline (see Figure 13a); hence, there should be *easy discovery* between the wireless sensor module on the robot and the receivers. Moreover, we need a wireless system that *can penetrate the harsh underground environment* with high signal attenuation [16] and it is *bidirectional* to facilitate sensor measurement transmission from the robot to the base station, and the motion controls the command transmission from the base station to the robot.

We propose a wireless sensor system based on an active low frequency (RFID) that facilitates easy discovery between the transceivers and can work in low frequencies for the better penetration of harsh environments.

### 5.1. Bi-Directional Wireless Sensor Module

We design a wireless sensor module based on the proposed active RFID wireless system using CC1200 from Texas Instruments (TI) Inc© as a physical layer. It can **receive and send data** and has 128 bytes data FIFO for both TX and RX sides. The functions of CC1200 are controlled by the host MCU. Up to five sensors are connected to five ADC channels of the MCU and the payload of the CC1200 is updated with processed sensors measurements in each transmission.

Next, we configure the CC1220 registers to ensure a reliable communication link with maximum throughput. The register configuration is facilitated by the SmartRF tool that derives the proper value for registers based on the defined features. We configured the radio signal with 434 MHz carrier frequency for underground environment penetration and long read range for the signal. Moreover, to receive the maximum possible signal strength at the receiver side, we configured the maximum transmission power that CC1200 can support (i.e., 14 dBm) and to have the maximum throughput, we configured the symbol rate of 1250 symbols per second (sps), and the modulation format of the 4-Gaussian frequency shift keying (4-GFSK) that is the maximum throughput of CC1200.

### 5.2. Experiments for the Wireless Sensor Module

We proved the bi-directionality of the wireless sensor module and also the fast connection capabilities. We set an experiment to measure the received signal strength (RSS) of the transmitted signal of our wireless sensor module, and the packet error (i.e., the number of packets that are received incorrectly). However, the characteristic of underground environment is variable in different locations. The properties of soil (i.e., water content, temperature, soil bulk density, sand/clay composition) are highly variable and each parameter has a specific impact on the communication channel. For example, the increase in water content of soil increases signal attenuation, while the increase in the sand composition of soil decreases the wireless signal attenuation. Hence, the experiments in one location in soil **cannot be generalized**. To this aim, we need to provide a solution to measure the read range of the wireless sensor module and validate it with experiment. The researchers in [16] simulated the signal attenuation per meter in the soil with varying parameters (i.e., sand, clay, water, the carrier frequency). In one scenario, the volumetric water content of soil and carrier frequency remains constant in 5% and 2.4 GHz, and the sand and clay composition of soil changes. In this scenario, the signal attenuation varies from 10 dB to 90 dB for different compositions of sand, clay, and silt. In another scenario, carrier frequency and volumetric water contents vary (i.e., 0–25%), and soil content remains constant (i.e., 50% sand, 15% clay, and 35% silt). In this scenario, the signal attenuation per meter changes from around 40 dB to 140 dB in the 2.4 GHz carrier frequency and volumetric water contents from 0–25%, respectively, and is higher than the signal attenuation per meter for the varying sand/clay composition in rather dry soil (i.e., 5% volumetric water content). Hence, the effect of volumetric water content on signal attenuation is dominant compared to other parameters, and we can employ this fact to provide the minimum read range of our wireless sensor module in the underground environment.

To this aim, we did our experiments in a water environment that has a constant characteristic and designed a setup, as shown in Figure 13b. The setup comprises five FlexiForce sensors that are connected to the wireless system and both are powered by a battery. The wireless sensor module is immersed in a water pool and sends the sensor measurements, continuously (Figure 13c). The received signal strength indicator (RSSI) module (TrxEB from TI) is located above-ground. In Figure 13d, *D* is the vertical distance (i.e., in water) of the wireless sensor module and RSSI, and *H* is the horizontal distance between them in air. We measured RSS for different values for *D* (i.e., 10 cm, 20 cm, 40 cm, and 100 cm) and for each *D*, 16 measurements are conducted for different values for *H*, ranging from 0 m to 15 m (see Figure 13c). In *D* = 10 cm, RSS is −66 dBm and fluctuates around this amount with small changes in different values for *H*. In *D* = 20 cm, RSS is −73 dBm at *H*≈ 0 and fluctuates with small changes in different horizontal distances. At *D* = 40 cm, RSS is measured −80 dBm and does not change considerably at different horizontal distances. Moreover, in *D* = 100 cm, RSS has measured −100 dBm and the changes are small in different horizontal values. We conclude from the results that the distance between two transceivers in the air does not affect the RSS, and the read range of the wireless sensor module in the water environment is 1 m. Hence, we have a wireless sensor module that penetrates the harsh environment and can broadcast the data packet of the five sensor measurements in 1 m *if only* water that is the most attenuating environment is present with the ≈0% packet error. The SmartCrawler works in 9-in to 22-in diameter pipe and the rest of the environment is soil, so the read range is more than 1 m. Based on the standards in [23], the pipelines are buried in 1.5 m beneath the earth; considering the maximum allowable pipe diameter for our robot (i.e., 0.56 cm), the RSS is around −80 dBm and, given −100 dBm for the correct realization of data packets, extra −20 dBm (i.e., −50 dB) path loss is possible. The allowable path loss matches the results in [16] for path loss per meter in 2.4 GHz.

## 6. Integrated Test

We synchronize the wireless system with the motion controller and evaluate its performance in three iterations. To this aim, we propose a relay node comprising two CC1200 evaluation modules (one for motion command transmission and one for sensors’ measurements transmission) with carrier frequency 434 MHz and are connected to two MCU launchpads in Figure 14a. The relay node sends a motion command (so-called “start-motion”) and initializes the robot motion. The measurements of five sensors are transmitted to the relay nodes continuously during motion of the robot and the results are shown in a GUI (see Figure 14b). The robot stops once the relay node sends **another** command, the so-called “stop-motion”. The sequences of the motion of the robot are shown in Figure 14c and the value of linear speed and orientation are shown in Figure 15a–c. In each iteration, the robot reaches the desired velocity and cancels the deviations of ϕ and ψ and moves with constant velocity and stabilized configuration. Once the stop-motion is received by the robot, it stops with the stabilized configuration. Hence, with the developed integrated system, the motion of the robot and wireless sensor transmission is facilitated for the SmartCrawler.

### Accurate Localization

The robot switches from phase 1 to phase 2 based on the wireless signal from outside pipe. However, during our experiments in the “integrated test”, we noticed that the robot always receives the radio signal ***before*** it reaches the junction. The difference between the robot distance from the front obstacle (i.e., pipe wall) and the junction diameter is referred to as the dead distance, $D_{dead}$ (see Figure 16 for detail), which makes the exact robot localization impossible. To address this challenge, we mounted an ultrasonic sensor (for distance measurement) in front of the robot and modified the firmware in a way that once the radio signal is received by the robot, $D_{dead}$ is measured and the robot continues in phase 1 until $D_{dead}\to 0$. Then, it switches to phase 2. This way, the inaccuracy of robot localization is addressed.

## 7. Discussion

The waterproof and self-powered robot with customized gear motors, battery, and spring provide full autonomy in pipe missions. We characterized spring stiffness based on *water flow condition* and *pipe-wheel friction coefficient in WDS*, which prevents the slippage of wheels.

The proposed two-phase motion controller provides reliable motion for the robot in straight and non-straight paths. For non-straight paths, the controller enables the robot to rotate the desired amount, around y or Z axes, with one set of actuator modules that leads to system simplicity. T-junctions’ negotiation is a big challenge for in-pipe robots, especially in *large pipes* of WDS, and is not discussed in the literature [7,24]. Based on our simulations, the key parameter in the reliability of the controller in phase 2 (specifically at T-junctions) is **pretension in the springs**. Slippage does not occur during rotation and the deviations in ϕ and ψ are canceled at the end of rotation. Given the high accuracy of the selected observer sensors, and the fluctuation bound for the stabilizing states, this is not a concern in our application. We also defined the size limitation for the rotation of the robot in bends and T-junctions to meet the pretension requirement for the springs. As for the drift in observer sensors, the selected observer sensors are selected from precise applications. Moreover, in our experiments, we found out that with $\pm 2^{\circ }$ of fluctuation in stabilizing states, the robot can operate smoothly in the pipe.

The wireless sensor module that is proposed in this paper can be used for multi-parameter measurements and transmission of up to five parameters in water that shows improvement compared to mobile sensors [2]. Moreover, the bidirectional wireless module penetrates underground environments at higher read range [25], and its performance is not prone to *the orientation of the antenna* [26]. Increasing the number of physical layers requires additional antenna(s) that are extremely challenging to locate in the confined space of the central processor. Moreover, the high capacity of Data FIFO in each TX and RX side (128 bytes for each) in one CC1200 provides sufficient memory for payload updating for five sensor measurements.

The integration of the wireless sensor module and two-phase motion controller offers promising results for the in-pipe robots to access them and exchange data during operation, and the developed wireless control mechanism facilitates long-distance inspection for the in-pipe missions. The localization inaccuracy is also addressed using an ultrasonic sensor in front of the system.

## 8. Conclusions and Future Works

In this research, we designed and developed an integrated in-pipe wireless robotic system in distribution systems. The modular architecture of the robot with customized elements based on water flow conditions provide full autonomy for the robot in in-service WDS. Moreover, the synchronized two-phase motion controller and the wireless sensor module provides reliable motion in complicated configurations of WDS and wireless data transmission in underground environments where the signal attenuation is high.

In our future work, we will develop an operation procedure that facilitates smart navigation and data transmission during the operation for the SmartCrawler. To this aim, some radio transceivers are placed above-ground and the robot moves underground inside the pipeline (see Figure 17). The robot switches its wireless communication between different transceivers. **Map of the operation** is given to the robot as a matrix, to give it the perception of the configuration type of non-straight configurations. Based on our experiments, the above-ground relay node and the underground robot can communicate with each other once they are in the range of each other.

## Figures and Tables

**Figure 1 sensors-22-09666-f001:**
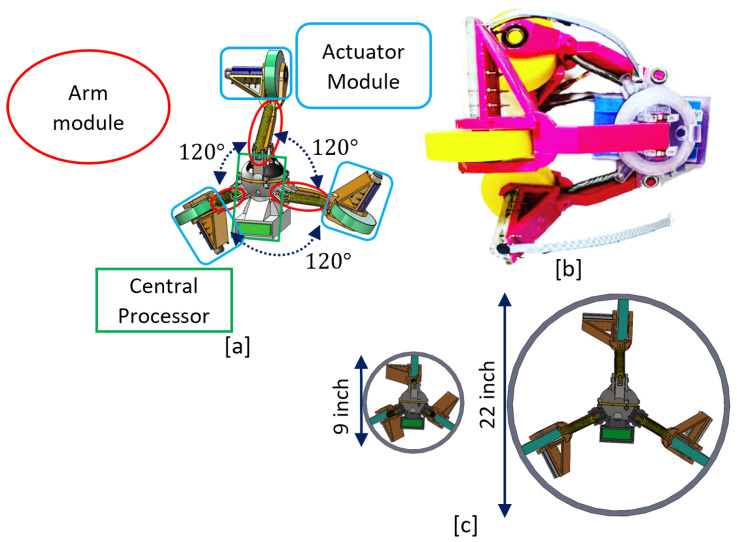
SmartCrawler. (**a**) CAD Design. (**b**) Prototype. (**c**) Minimum and maximum size (Figures are on scale).

**Figure 2 sensors-22-09666-f002:**
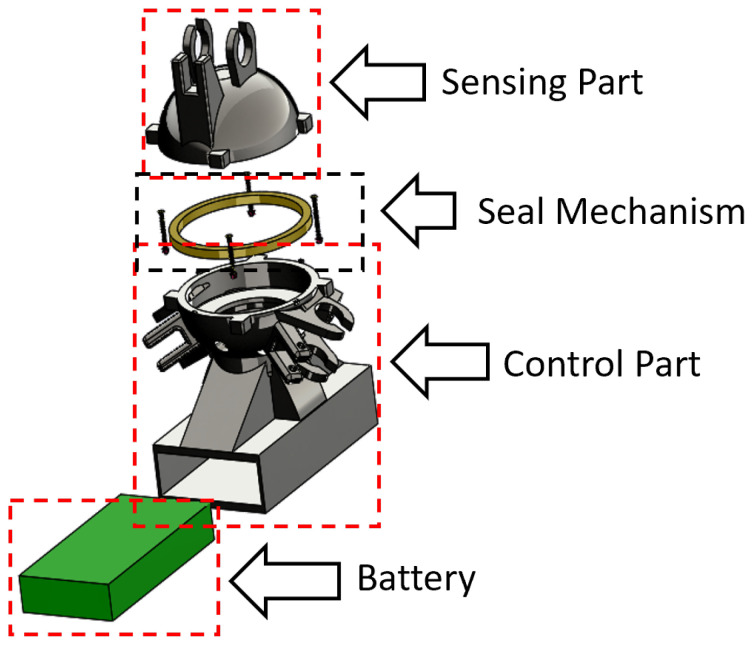
The CAD design of the central processor.

**Figure 3 sensors-22-09666-f003:**
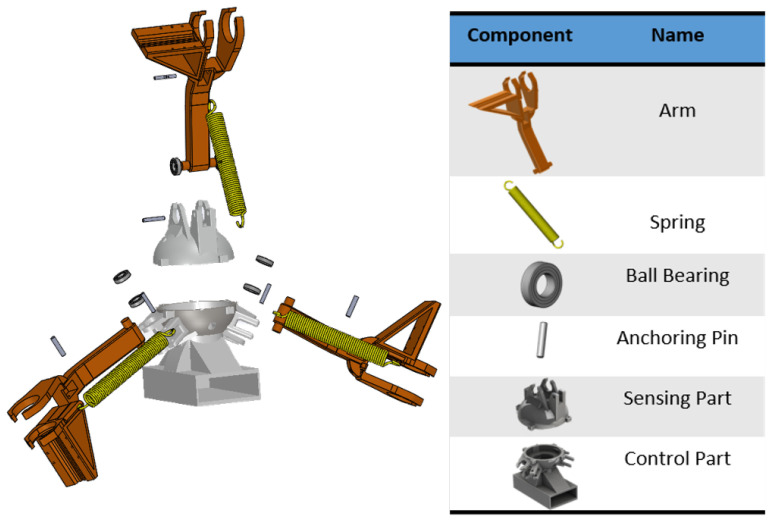
The CAD design of the arm Module.

**Figure 4 sensors-22-09666-f004:**
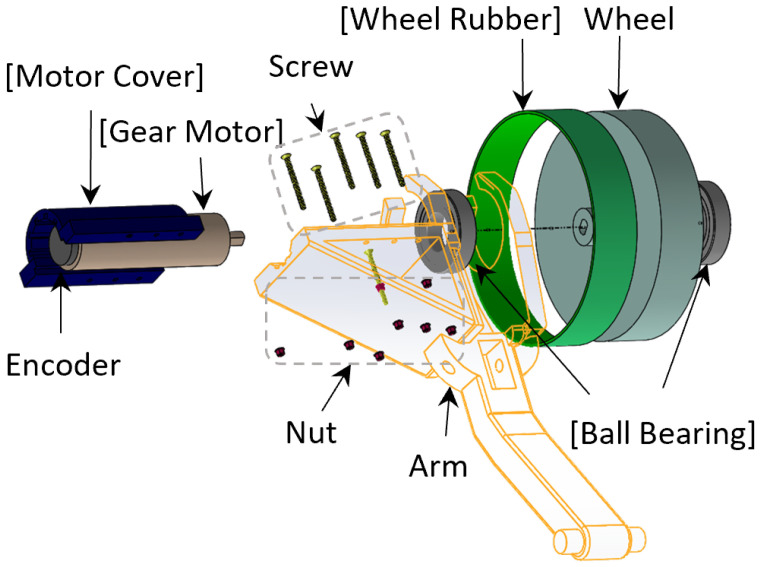
The CAD design of the actuator module.

**Figure 5 sensors-22-09666-f005:**
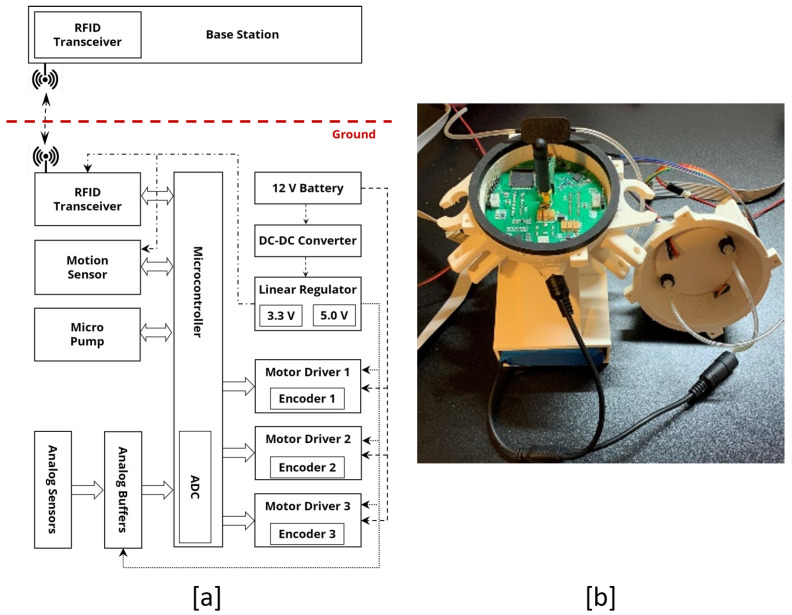
(**a**) Hardware architecture of PCB of the SmartCrawler. (**b**) PCB of the robot and the mechanical integration with the central processor.

**Figure 6 sensors-22-09666-f006:**
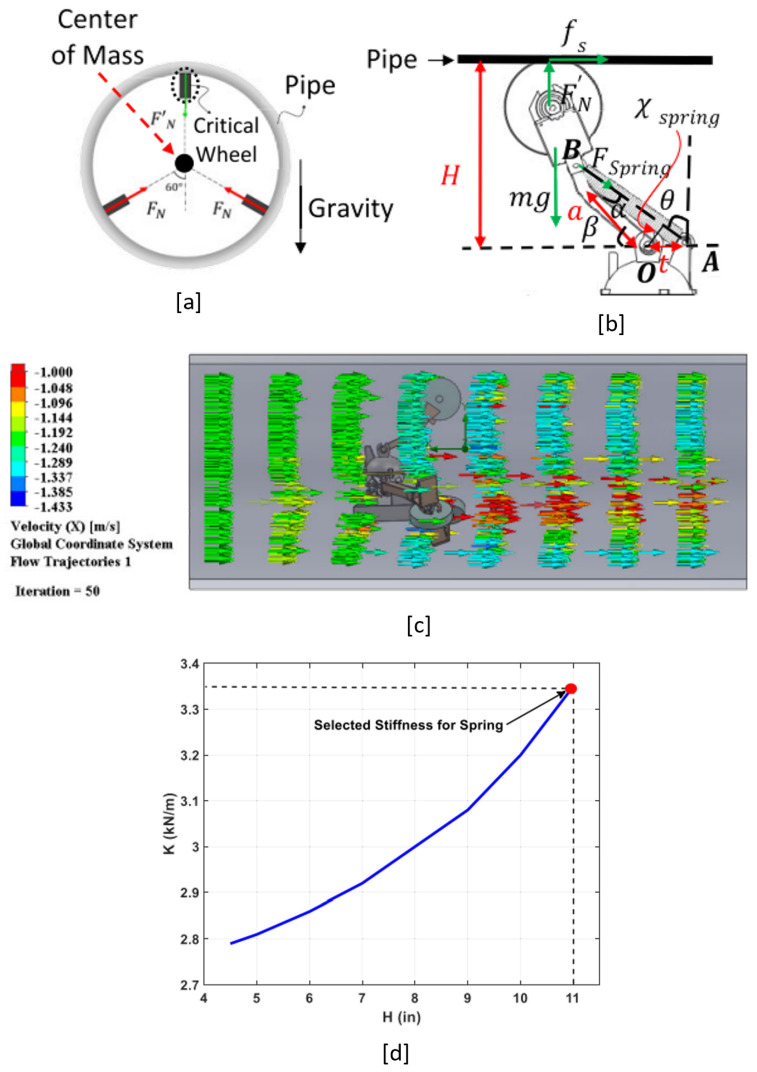
(**a**) Normal forces between the wheels and the pipe wall. (**b**) Static force analysis for the critical wheel in (**a**). (**c**) Flow simulation with CFD work in SolidWorks. (**d**) The required spring stiffness in different pipe sizes, *H*.

**Figure 7 sensors-22-09666-f007:**
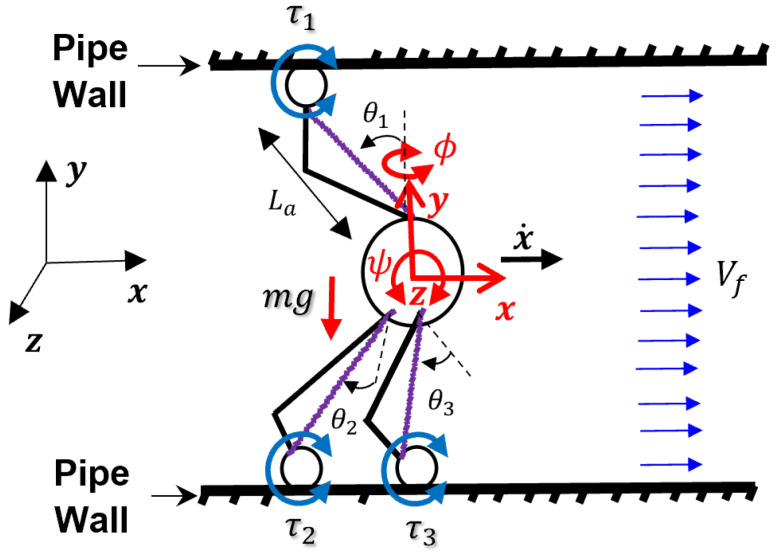
Free body diagram of the SmartCrawler.

**Figure 8 sensors-22-09666-f008:**
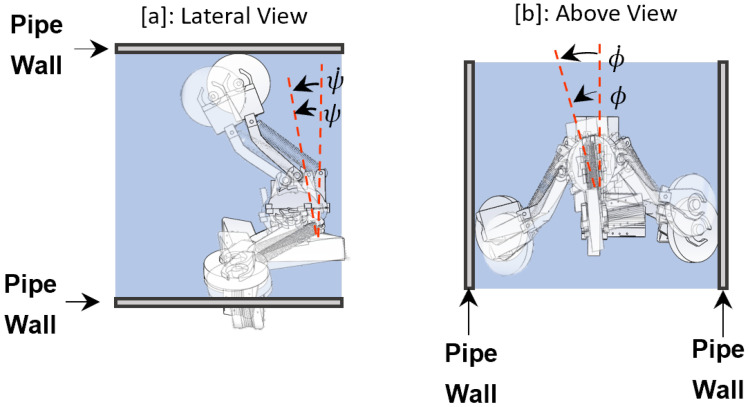
Stabilizing states of the SmartCrawler.

**Figure 9 sensors-22-09666-f009:**
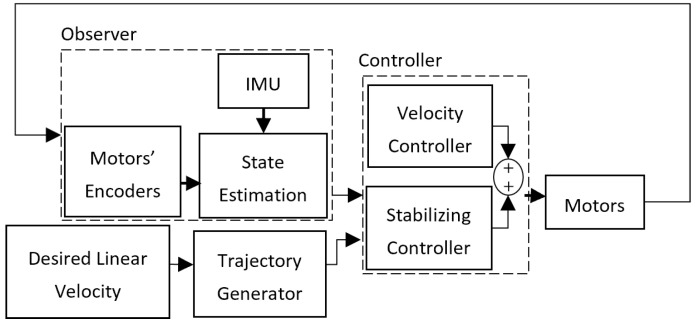
Stabilizer–velocity controller.

**Figure 10 sensors-22-09666-f010:**
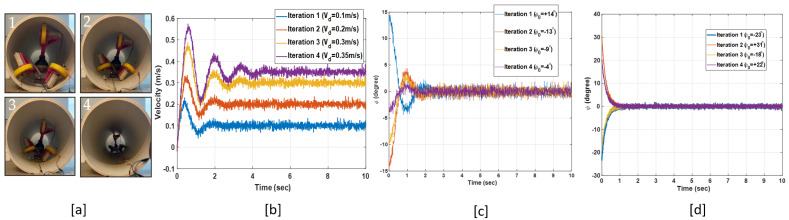
(**a**) Motion sequences of the robot. (**b**) Linear velocity (m/s). $v_{d}$: Desired linear velocity. (**c**) ϕ (Degree). (**d**) ψ (Degree).

**Figure 11 sensors-22-09666-f011:**
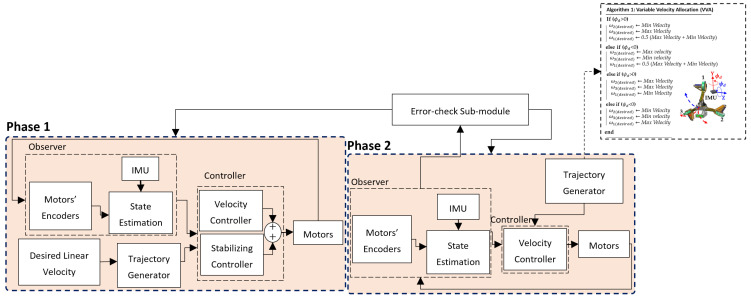
The two-phase motion controller of the SmartCrawler.

**Figure 12 sensors-22-09666-f012:**
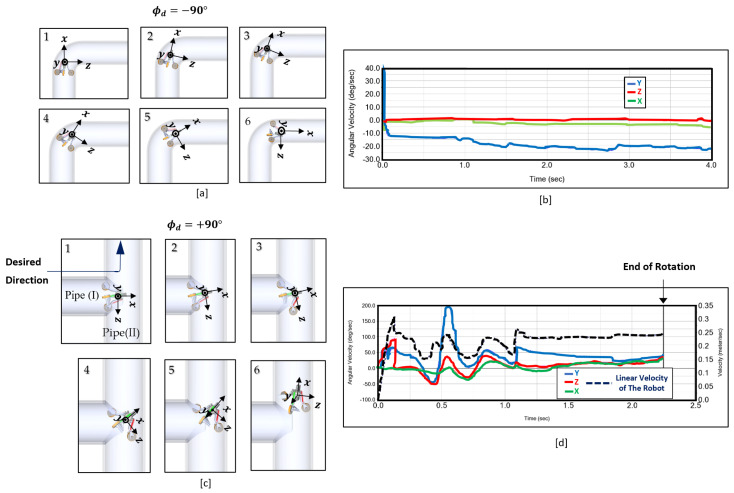
(**a**) The sequence of motion at 12-in bend. (**b**) The robot’s angular velocities at 12-in bend. (**c**) The sequence of motion at 12-in T-junction. (**d**) The robot’s angular velocities at 12-in T-junction.

**Figure 13 sensors-22-09666-f013:**
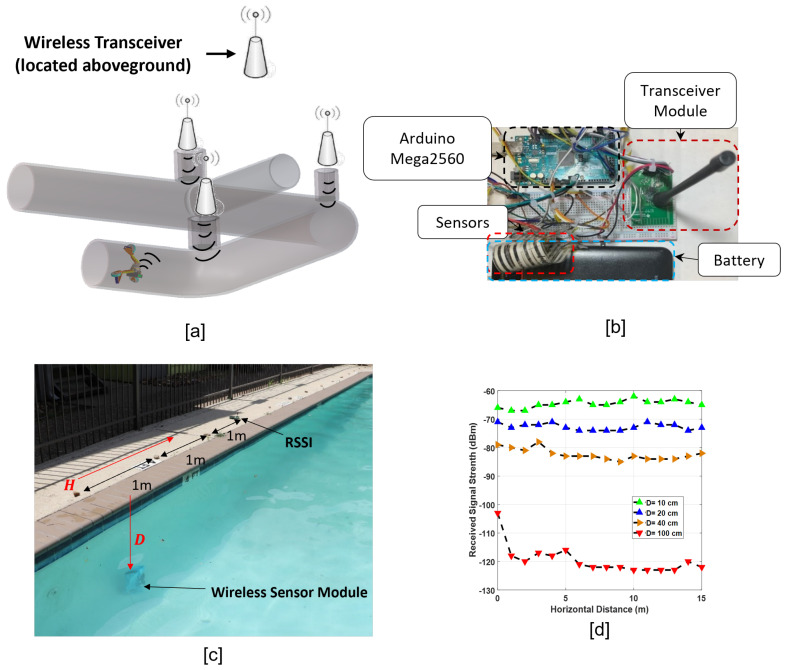
(**a**) The proposed architecture of the wireless communication system in our application. The transceiver(s) are located outside pipelines (**b**) Wireless sensor module. (**c**) The experiment setup. (**d**) Received signal strength (RSS) at different values of *D* (distance of transceivers in water) and *H* (distance of transceivers in air).

**Figure 14 sensors-22-09666-f014:**
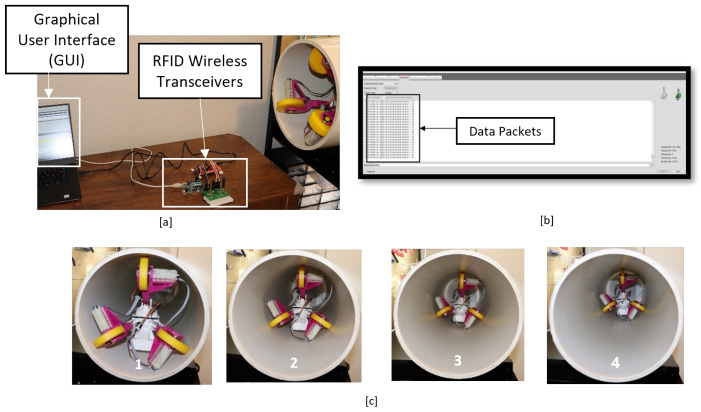
(**a**) Relay node and the robot inside pipe. (**b**) The received sensors measurements by GUI. (**c**) Sequences of motion.

**Figure 15 sensors-22-09666-f015:**
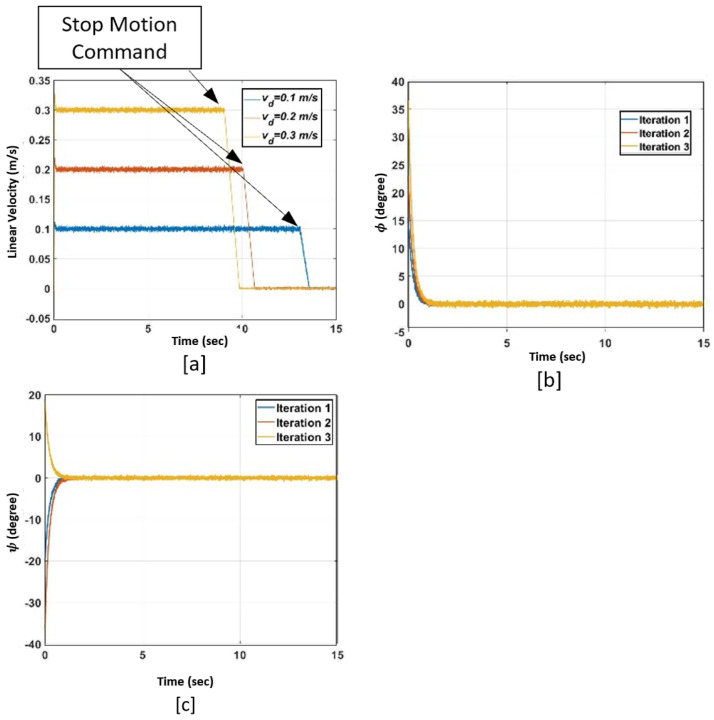
(**a**) Linear velocity. (**b**) Rotation around Y-axis (ϕ). (**c**) Rotation around Z-axis (ψ).

**Figure 16 sensors-22-09666-f016:**
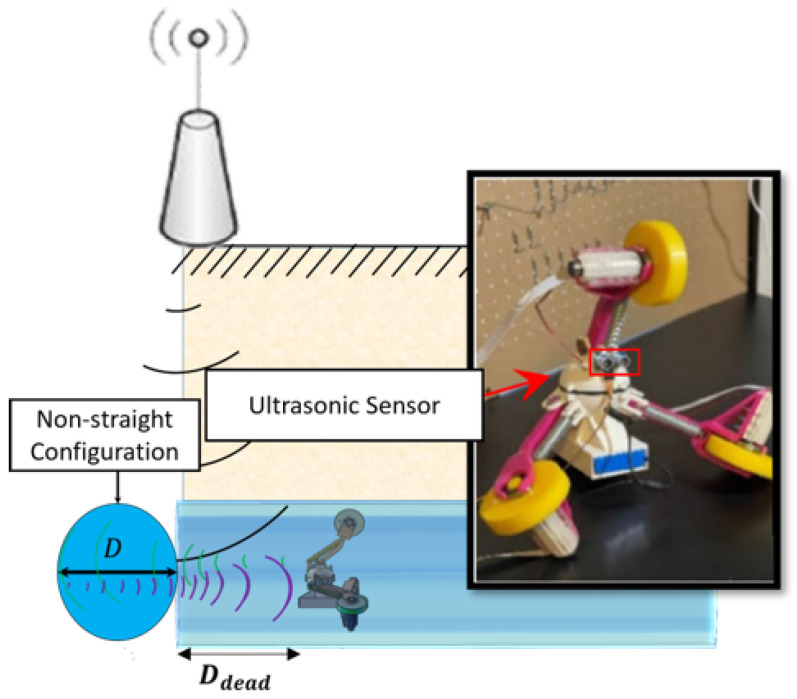
The definition of dead distance ($D_{dead}$) and the use of ultrasonic sensor for exact robot localization.

**Figure 17 sensors-22-09666-f017:**
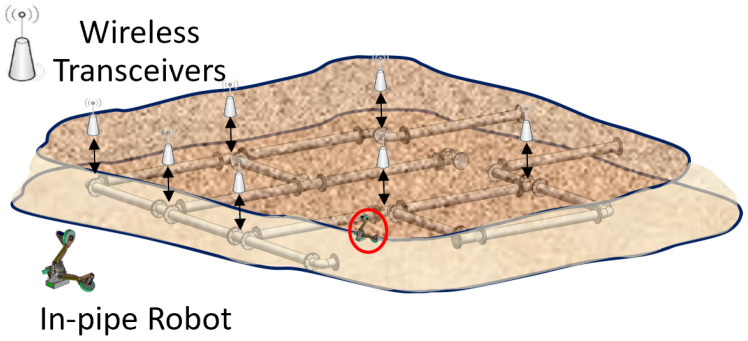
The proposed wireless sensor networks. Multiple wireless sensor transceivers are placed at non-straight configurations of pipeline and the robot switches its communication with different wireless transceivers.

**Table 1 sensors-22-09666-t001:** SmartCrawler specifications.

Parameter [Unit]	Description
Size Adaptability Range [in]	9–22
Length [in]	7.5–9
Maximum Linear Speed [m/s]	1.56
Vertical Motion Capability	Yes
Weight [kg]	2.23

**Table 2 sensors-22-09666-t002:** Parameters in SmartCrawler modeling.

Parameter [Unit]	Description	Value
$L_{a}$ [cm]	Arm Length	17
*R* [cm]	Wheel Radius	5
*m* [kg]	Robot Mass	2.23
$I_{yy}$ [kg·m${}^{2}$]	Moment of inertia of Y-axis	0.0126
$I_{zz}$ [kg·m${}^{2}$]	Moment of inertia of Z-axis	0.0093
ρ [kg/m${}^{3}$]	Water Density	1000
$C_{d}$	Drag Coefficient (computed with CFD)	0.47
$\tau_{1}$ [N·m]	Torque for actuator #1	Variable
$\tau_{2}$ [N·m]	Torque for actuator #2	Variable
$\tau_{3}$ [N·m]	Torque for actuator #3	Variable
*A*	Frontal area of the robot facing water	Variable
$\theta_{1}$ [${}^{\circ }$]	Arm #1 Angle	Variable
$\theta_{2}$ [${}^{\circ }$]	Arm #2 Angle	Variable
$\theta_{3}$ [${}^{\circ }$]	Arm #3 Angle	Variable
ϕ [${}^{\circ }$]	Rotation around y-axis	Variable
ψ [${}^{\circ }$]	Rotation around z-axis	Variable
$V_{f}$ [m/s]	Flow Velocity	Variable
$\dot{x}$ [m/s]	Robot Velocity	Variable

## Data Availability

Not applicable.

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
