# Peer review of "SmartCrawler: A Size-Adaptable In-Pipe Wireless Robotic System with Two-Phase Motion Control Algorithm in Water Distribution Systems†"

_sensors, 2022, doi:10.3390/s22249666_

Round 1

Reviewer 1 Report

This paper proposed a size-adaptable in-pipe wireless robotic system for water distribution systems. A motion control algorithm was developed to enable straight and rotation motion in the pipe. And a bi-directional wireless sensor module was proposed to enable wireless underground communication. However, the authors didn’t clarify the originality or novelty of the proposed method. There are some suggestions for authors:

1. Describe the novelty characteristics of the proposed two-phase motion control algorithm. The velocity controller of the SmartCrawler in straight is a common PID controller, so the Figure 4 is not unnecessary. It’s should consider how to realize stable control in highly pressurized in-service WDS as mentioned in 2.4. Technical Gap.

2. This paper chooses CC1220 to build the multi-receiver wireless network and do some experiments to test the effect. This method is lack of novelty.

Reviewer 2 Report

This paper presents a novel robotics system design for in-pipe inspection, including the necessary control systems to navigate such environments. A full system integration is presented including sensory payload and communication package, which can operate in water ducts during operation, making it particularly interesting.

The section on related work is rather short, I presume due to the shortage of operational robotics systems for in-pipe inspection. There are however other research communities looking into this subject, for instance, there are multiple soft robotics systems that have attempted to address these challenges.

On the design section, Fig1 could do with some more information on the arm module (where are the actuators, encoders, etc), and including the frame of reference as in Fig2 would also be an improvement. This would help, for example, making it more clear why only t3 and t2 contribute for eq2. Also, it is not clear until the end of the paper if the theta values are controlled with separate motors or if they are pre-tensioned using springs.

The control section is clear and thorough.

The test shown is quite limited, it would be good to see a demonstration of the two-phase controller proposed for the robot. It seems form the discussion, however, that such tests were performed. Even if these were limited tests and there is no relevant data to include in the paper, I encourage the authors to qualitatively address these tests in the discussion section. Figure 11 should be completely redone; it is very low quality, and the plots are illegible. Spring stiffness is mentioned in the discussion section as being characterised; however, the simulations section shows no data on this. If this is a relevant parameter, I would be interested in seeing how it influences performance in phase 2.

The discussion section includes a lot of new information. For example, the IMU, and type of encoders used. This should be presented in the design section. Instead, I’d like to hear more of the author’s thoughts on the system’s performance.

Writing needs some revision, words such as huge should be avoided. All figures could do with improvements.

In summary, I think this new robot design for in-pipe inspection, dedicated control system and implementation of the data forwarding are sounds contributions to the field and should be published. The paper lacks, however, on how the results are presented. The system design can be consolidated with information that is only presented towards the end of the paper. The influence of parameters such as the arm spring stiffness or pre-tension on the controller for the T junction would also considerably enrich the paper. A clearer results section is also necessary to demonstrate the ideas here presented.

Round 2

Reviewer 1 Report

This is a well revised paper and can be accepted.

Reviewer 2 Report

The grammar on the new section "Spring Characterization" needs a minor revision.

The manuscript was thoroughly revised and addressed all points. Best wishes to the authors.